# Search for Extreme Mass Ratio Inspirals Using Particle Swarm Optimization and Reduced Dimensionality Likelihoods

Xiao-Bo Zou [1,2,3,4], Soumya D. Mohanty [2,5,*], Hong-Gang Luo [1,4] and Yu-Xiao Liu [1,3,4]

1  School of Physical Science and Technology, Lanzhou University, Lanzhou 730000, China;
   zouxb18@lzu.edu.cn (X.-B.Z.); luohg@lzu.edu.cn (H.-G.L.); liuyx@lzu.edu.cn (Y.-X.L.)
2  Morningside Center of Mathematics, Academy of Mathematics and System Science, Chinese Academy of
   Sciences, 55, Zhong Guan Cun Donglu, Beijing 100190, China
3  Institute of Theoretical Physics & Research Center of Gravitation, Lanzhou University, Lanzhou 730000, China
4  Lanzhou Center for Theoretical Physics and Key Laboratory of Theoretical Physics of Gansu Province,
   Lanzhou University, Lanzhou 730000, China
5  Department of Physics and Astronomy, The University of Texas Rio Grande Valley,
   Brownsville, TX 78520, USA
*  Correspondence: soumya.mohanty@utrgv.edu

**Abstract:** Extreme-mass-ratio inspirals (EMRIs) are significant observational targets for space-borne gravitational wave detectors, namely, LISA, Taiji, and Tianqin, which involve the inspiral of stellar-mass compact objects into massive black holes (MBHs) with a mass range of approximately $10^4 \sim 10^7 M_\odot$. EMRIs are estimated to produce long-lived gravitational wave signals with more than $10^5$ cycles before plunge, making them an ideal laboratory for exploring the strong-gravity properties of the spacetimes around the MBHs, stellar dynamics in galactic nuclei, and properties of the MBHs itself. However, the complexity of the waveform model, which involves the superposition of multiple harmonics, as well as the high-dimensional and large-volume parameter space, make the fully coherent search challenging. In our previous work, we proposed a 10-dimensional search using Particle Swarm Optimization (PSO) with local maximization over the three initial angles. In this study, we extend the search to an 8-dimensional PSO with local maximization over both the three initial angles and the angles of spin direction of the MBH, where the latter contribute a time-independent amplitude to the waveforms. Additionally, we propose a 7-dimensional PSO search by using a fiducial value for the initial orbital frequency and shifting the corresponding 8-dimensional Time Delay Interferometry responses until a certain lag returns the corresponding 8-dimensional log-likelihood ratio's maximum. The reduced dimensionality likelihoods enable us to successfully search for EMRI signals with a duration of 0.5 years and signal-to-noise ratio of 50 within a wider search range than our previous study. However, the ranges used by both the LISA Data Challenge (LDC) and Mock LISA Data Challenge (MLDC) to generate their simulated signals are still wider than the those we currently employ in our direct searches. Consequently, we discuss further developments, such as using a hierarchical search to narrow down the search ranges of certain parameters and applying Graphics Processing Units to speed up the code. These advances aim to improve the efficiency, accuracy, and generality of the EMRI search algorithm.

**Keywords:** LISA; Gravitational waves; EMRI; PSO; Likelihood ratio

## 1. Introduction

The extreme-mass-ratio inspirals (EMRIs) are sources of gravitational waves (GWs), where stellar-mass compact objects (COs) are captured and spiral into massive black holes (MBHs) at the centers of galaxies [1–3]. The emission of GWs gradually causes the eccentric orbit to shrink and become more circular. During the last year of inspirals before plunge, it is estimated that over $10^5$ cycles can be observed by space-based GW observatories [4], such as Taiji [5,6], Tianqin [7], and LISA [8]. The rich information from the phase evolution can

be utilized to constrain gravity theories beyond general relativity [9–11]; test the no-hair theorem [12–14], deviations from the Kerr metric [15–17], and dark matter [18–20]; and study the astrophysics of galaxies [21–23] with a high precision. The event rates detectable by LISA or Tianqin can vary from dozens to thousands, depending on the population models [2–4]. Consequently, a catalog of EMRI sources could potentially serve as dark sirens to constrain cosmological parameters, particularly the Hubble constant [24,25]. Therefore, EMRI data analysis becomes a crucial task.

Time-frequency methods provide a straightforward solution for detecting high SNR signals without the need for waveform models. Once the signal tracks in the time-frequency plane are well fitted, the waveform models can be used to estimate a subset of source parameters [26,27]. The advantage of this approach is that it is computationally cheap. However, the disadvantage is that it requires a lot of tuning for the threshold fitting in the time–frequency plane, and it is difficult to detect signals with a low SNR. Recently, Convolutional Neural Network (CNN)-based methods have been developed, where different inputs such as time domain data [28], frequency domain data [29], and time–frequency planes by Q-transform [30,31] are fed to the neural network. These methods provide an alternative computationally efficient solution for EMRI data analysis, but they are still limited to high SNR signals.

Template-based matched filtering is the best option for a deeper search in SNRs, although it is computationally expensive. In EMRI data analysis, accurate EMRI waveforms are quite complicated and computationally expensive when considering the self-force of the COs [32]. As a result, phenomenological waveforms from the kluge family are widely used at present in the development of EMRI data analysis methods. The analytical kluge (AK) waveform [33] is used in the Mock LISA Data Challenges (MLDCs) [34–37] and the latest LISA Data Challenge (LDC) [38], while the augmented analytical kluge (AAK) waveform [39] is used in the Taiji data challenge [40]. The AK waveform includes 14 parameters, with the spin of the COs usually being ignored. Six of these parameters contribute to the phase evolution of the waveform and need to be estimated with high precision, thus contributing a prominent 6-dimensional sharp peak to the signal location in the parameter space of the fitness function. The AK waveform consists of a superposition of multiple harmonics, resulting in multiple secondary peaks surrounding the primary one in the parameter space [41]. The primary peak indicates a good match of all the harmonics, while the secondary peaks indicate that a subset of harmonics is matched well, especially the dominant ones. Therefore, it is difficult for a global optimizer to locate a complete signal in such a high-dimensional and multimodal parameter space.

It is well known that longer-duration signals contribute more sensitivity and less flexibility to coherent matched filtering [42], and the sharp peak can only be located within a reasonable range width [43]. As a result, hierarchical search methods are effective in overcoming the methodological difficulties in EMRI data analysis. It can be implemented by either using shorter-duration signals and gradually turning to longer signals with the constrained information utilized in the next search [42,44] or by initially searching for fixed-duration signal within a wide range and later focusing on narrower ranges extracted from the previous searches [43] in matched filtering. It is also beneficial to develop mixed versions by combining these two approaches together.

Given the fitness function usually defined by the log-likelihood ratio (LLR), Bayesian [45] or Fisherian methods [46] are the most commonly used ones for estimating the posterior probability density function or the global best-fit fitness value and location, which are then used for signal detection and parameter estimation. In EMRI data analysis, modified Markov Chain Monte Carlo (MCMC) methods, such as constrained Metropolis–Hastings Monte Carlo (MHMC) [44], Evolutionary Monte Carlo (EMC) [47], and parallel tempered Markov Chain Monte Carlo (PTMCMC) [48,49], have been used in previous works. The global optimizer, Particle Swarm Optimization (PSO), first proposed in [50,51] and validated by [52,53], was used in our previous work for a 10-dimensional EMRI search problem [54] and was proven effective in the LIGO data analysis of inspiral signals [55–58] and transient

signals [59–62], the pulsar timing array data analysis of supermassive black holes [63–68], and the LISA data analysis of galactic binaries [69–73]. Besides gravitational wave data analysis, it is also widely used in other fields, such as electromagnetics [74,75]. More comprehensive discussions can be seen in [76–79]. The advantages of PSO are its fewer tunable parameters and smaller number of LLR evaluations needed to reach the global LLR maximum compared with the MCMC method. In this paper, we extend the application of PSO to an EMRI search problem with two different dimensions: an 8-dimensional search and a 7-dimensional search, respectively. Our results demonstrate that the PSO-based search algorithm is able to accurately estimate the simulated signals with an SNR value of 50 and a duration of 0.5 years by using these reduced dimensionality LLRs. Notably, it should be emphasized that the current search ranges employed are substantially broader than those utilized in our previous work, resulting in a significant increase in the parameter space volume, approximately ∼50-fold for the 8-dimensional search and ∼100-fold for the 7-dimensional search.

The rest of the paper is organized as follows. In Section 2, we describe the consistent TDI combinations, noise model, and the signal model as LDCs used in the paper. In Section 3, we present how the reduced dimensionality likelihoods are defined. The Particle Swarm Optimization algorithm used for matched filtering is illustrated in Section 4. Finally, in Section 5, we report the results and give the corresponding discussions in Section 6.

## 2. Data Description

First, we describe the application of time-delay interferometry (TDI) [80] in this paper, which is employed by space-based GW detectors to mitigate the dominant laser frequency noise. Subsequently, we present the theoretical model of power spectral densities (PSDs) utilized by LDCs. Lastly, we provide a description of the current standard waveform model employed for EMRI data analysis.

### 2.1. TDI Combinations

Throughout the paper, we adhere to the coordinate and TDI conventions defined in [81]. Given the definitions of the polarization tensors $\epsilon^{+,\times}$ and LISA orbit, we can derive the corresponding geometrical quantities $\widehat{n}_l$ and $\widehat{R}_k$ from the orbit. Here, $\widehat{n}_l$ represents the unit vector along the arm link $l$ between the two involved satellites, and $\widehat{R}_k$ denotes the position vector of the $k$-th satellite. The sky's location, $\theta_s$ and $\phi_s$, can be used to define the unit vector $\hat{k}$ which indicates the direction of the GW propagation. The antenna patterns $F_l^{+,\times}$ of the single arm $l$ are given by

$$\begin{bmatrix} F_l^+ \\ F_l^\times \end{bmatrix} = \begin{bmatrix} \cos(2\psi) & -\sin(2\psi) \\ \sin(2\psi) & \cos(2\psi) \end{bmatrix} \begin{bmatrix} U_l^+ \\ U_l^\times \end{bmatrix}, \tag{1}$$

where $\psi$ is the polarization angle and the quantities $U_l^{+,\times}$ are defined by

$$U_l^+ = (\widehat{n}_l \otimes \widehat{n}_l) : \epsilon^+ , \tag{2}$$

$$U_l^\times = (\widehat{n}_l \otimes \widehat{n}_l) : \epsilon^\times . \tag{3}$$

The symbol : denotes the contraction operation on arbitrary tensors $U$ and $V$, namely, $U : V = \sum_{i,j} U_{ij} V_{ij}$, and $\otimes$ represents $(a \otimes b)_{ij} = a_i b_j$ for arbitrary vectors $a$ and $b$.

By mapping the antenna patterns $F_l^{+,\times}$ to the polarized waveforms $h_{+,\times}$, we can express the corresponding strain response of the arm $l$ as $\Phi_l$:

$$\Phi_l = F_l^+ h_+ + F_l^\times h_\times . \tag{4}$$

The expression for the single-arm response of the laser along the arm $l$ can then be given as follows:

$$y_{slr}^{\text{GW}}(t) = \frac{\Phi_l(t - \hat{k} \cdot \hat{R}_s - L_l) - \Phi_l(t - \hat{k} \cdot \hat{R}_r)}{2(1 - \hat{k} \cdot \hat{n}_l)} \, , \tag{5}$$

where the labels $s$ and $r$ represent the laser sender and receiver of the satellite, respectively, and $L_l$ is the corresponding arm length of link $l$. The sign of $l$ is positive when the label $slr$ follows a cyclic permutation of indices $1 \rightarrow 2 \rightarrow 3 \rightarrow 1$ labeling the three satellites; otherwise, it is negative. By following the well-designed optical path of the TDI combinations $X$, $Y$, and $Z$ of the first generation, the laser frequency noise can be canceled under the approximation of a constant arm length. This cancellation is achieved by linearly combining the artificially delayed single-arm responses $y_{slr,L_l}$, as shown below:

$$
\begin{aligned}
X &= y_{1-32,32-2} + y_{231,2-2} + y_{123,-2} + y_{3-21} - y_{123,-2-33} - y_{3-21,-33} - y_{1-32,3} - y_{231} \, , \\
Y &= y_{2-13,13-3} + y_{312,3-3} + y_{231,-3} + y_{1-32} - y_{231,-3-11} - y_{1-32,-22} - y_{2-13,1} - y_{312} \, , \quad (6) \\
Z &= y_{3-21,21-1} + y_{123,1-1} + y_{312,-1} + y_{2-13} - y_{312,-1-22} - y_{2-13,-11} - y_{3-21,2} - y_{123} \, ,
\end{aligned}
$$

where the $y_{slr,L_l}$ are linked to the single-arm responses $y_{slr}$ through $y_{slr,L_l}(t) = y_{slr}(t - L_l)$. The first-generation TDI combination $X$ (same for $Y$ or $Z$) is calculated as the difference between two Michelson-type responses. Each Michelson-type response consists of an optical path with 4 single-arm responses. Each single-arm response introduces a delay corresponding to its arm length along the optical path. Consequently, there are 0, 1, 2, and 3 accumulated indices for the delays $L_l$ in the $y_{slr,L_l}$ of Equation (6), respectively, and the corresponding signs follow the same rule as the links $l$. Additionally, we can obtain the mutually independent noise TDI combinations $A$, $E$, and $T$ by linearly combining the TDI combinations $X$, $Y$, and $Z$ as follows:

$$
\begin{aligned}
A &= \frac{Z - X}{\sqrt{2}} \, , \\
E &= \frac{X - 2Y + Z}{\sqrt{6}} \, , \quad (7) \\
T &= \frac{X + Y + Z}{\sqrt{3}} \, .
\end{aligned}
$$

In this paper, we focus on the data analysis method for an individual EMRI source. As a result, the corresponding data model of each combination $I$ is described by

$$\overline{d}^I = \overline{h}^I + \overline{n}^I \, , \tag{8}$$

where $\overline{d}^I$ represents the TDI combination $I$, with $I \in \{A, E, T\}$; $\overline{h}^I$ denotes the single EMRI signal; and $\overline{n}^I$ represents the purely instrumental noise for simplicity. We concentrate solely on the TDI combinations $A$ and $E$ because the TDI combination $T$ is less sensitive to GWs, which aligns with the treatment employed by numerous other studies.

*2.2. Noise Model and Signal-to-Noise Ratio*

We utilize the identical PSD model of TDI combinations $A$ and $E$ of the first generation, as provided in [81]:

$$S_n^A(f) = S_n^E(f) = S_n(f) = 8\sin^2\omega L \big[4(1 + \cos\omega L + \cos^2\omega L)S^{\text{Acc}} + (2 + \cos\omega L)S^{\text{IMS}}\big] \, , \tag{9}$$

where $\omega$ is the angular frequency of gravitational waves, $f = \frac{\omega}{2\pi}$ is the corresponding frequency in Hz, and $L$ is the constant arm length whose value is $2.5 \times 10^9$ m in the current

design of LISA. The acceleration noise $S^{\text{Acc}}$ and the Instrumental Optical Metrology System noise $S^{\text{IMS}}$ under the noise model "SciRDv1" are defined in [82] as follows:

$$
\begin{aligned}
S^{\text{Acc}}(f) &= \frac{9.0 \times 10^{-30}}{(2\pi f c)^2}\Big[1 + \big(\frac{0.4\text{mHz}}{f}\big)^2\Big]\Big[1 + \big(\frac{f}{8\text{mHz}}\big)^4\Big]\frac{1}{\text{Hz}} \ , \\
S^{\text{IMS}}(f) &= 2.25 \times 10^{-22}\big(\frac{2\pi f}{c}\big)^2\Big[1 + \big(\frac{2\text{mHz}}{f}\big)^4\Big]\frac{1}{\text{Hz}} \ .
\end{aligned}
\tag{10}
$$

Having acquired the analytical expressions of the PSD, the inner product between two signals $\bar{a}$ and $\bar{b}$ is defined by

$$
(\bar{a}|\bar{b}) = \frac{1}{Nf_s}\sum_{k=0}^{N-1}\frac{\widetilde{a}_k\widetilde{b}_k^* + \widetilde{a}_k^*\widetilde{b}_k}{S_n(f_k)} \ ,
\tag{11}
$$

where $\widetilde{x}$ denotes the Discrete Fourier Transform (DFT) of a time series $\bar{x} = (x_0, x_1, \ldots, x_{N-1})$,

$$
\begin{aligned}
\widetilde{x} &= \mathbf{F}\bar{x}^T \ , \tag{12} \\
F_{lm} &= e^{-2\pi i l m/N} \ , \tag{13}
\end{aligned}
$$

and $f_k = kf_s/N$, $k = 0, 1, \ldots, N-1$, with $f_s$ being the sampling frequency. In terms of the inner product, the SNR of a signal can be defined as follows:

$$
\text{SNR}^2 = (\bar{h}^A|\bar{h}^A) + (\bar{h}^E|\bar{h}^E) \ .
\tag{14}
$$

It is also convenient to define the combined overlap between two signals $\bar{h}_1^I$ and $\bar{h}_2^I$ ($I \in \{A, E\}$) as follows:

$$
\text{ff}_{\text{AE}} = \frac{(\bar{h}_1^A|\bar{h}_2^A) + (\bar{h}_1^E|\bar{h}_2^E)}{\sqrt{(\bar{h}_1^A|\bar{h}_1^A) + (\bar{h}_1^E|\bar{h}_1^E)}\sqrt{(\bar{h}_2^A|\bar{h}_2^A) + (\bar{h}_2^E|\bar{h}_2^E)}} \ ,
\tag{15}
$$

which is commonly used to assess the quality of the match between injected and estimated signals in mock data analysis [34]. The overlap of the individual combination, either $A$ or $E$, can be obtained by setting the other combination to zero.

### 2.3. Signal Model: EMRI Waveform

The AK waveform [33] includes 14 parameters, namely, $\mu$, $M$, $\lambda$, $S/M^2$, $e_0$, $\nu_0$, $\theta_s$, $\phi_s$, $\theta_k$, $\phi_k$, $\phi_0$, $\widetilde{\gamma}_0$, $\alpha_0$, and $D$. The first six parameters represent the mass of the COs, the mass of the MBH, the inclination angle between the orbital angular momentum of the COs and the spin direction of the MBH, the spin magnitude of the MBH, the initial orbital eccentricity, and the initial orbital frequency. These parameters contribute to the orbital dynamics of EMRI sources. The angles $\theta_s$ and $\phi_s$ denote the ecliptic colatitude and longitude of the source's sky location in the Solar System Barycenter (SSB) frame, while $\theta_k$ and $\phi_k$ represent the polar and azimuthal angles of the spin direction of the MBH in the SSB frame. Additionally, $\phi_0$, $\widetilde{\gamma}_0$, and $\alpha_0$ correspond to the initial angles of orbital motion, pericenter precession, and Lense–Thirring precession, respectively. Finally, $D$ represents the distance between the source and the SSB center. The polarization angle $\psi$ is a constant in the static frame, as discussed in [44], and depends on $\theta_s$, $\phi_s$, $\theta_k$, and $\phi_k$.

The orbital dynamics in the AK waveform are described by the following set of ordinary differential equations (ODEs). These ODEs contain five quantities: $\phi$, $\nu$, $\widetilde{\gamma}$, $e$, and $\alpha$, where $\nu$ and $e$ are the orbital frequency and the orbital eccentricity, respectively, and $\phi$, $\widetilde{\gamma}$, and $\alpha$ are the phases describing orbital motion, pericenter precession, and Lense–Thirring precession, respectively.

$$\frac{d\phi}{dt} = 2\pi\nu \, , \tag{16}$$

$$\begin{aligned}
\frac{d\nu}{dt} = {}& \frac{96}{10\pi}(\mu/M^3)(2\pi M\nu)^{11/3}(1-e^2)^{-9/2}\Big\{\Big[1+(73/24)e^2+(37/96)e^4\Big](1-e^2) \\
&+(2\pi M\nu)^{2/3}\Big[(1273/336)-(2561/224)e^2-(3885/128)e^4-(13{,}147/5376)e^6\Big] \\
&-(2\pi M\nu)(S/M^2)\cos\lambda(1-e^2)^{-1/2}\big[(73/12)+(1211/24)e^2 \\
&+(3143/96)e^4+(65/64)e^6\big]\Big\} \, ,
\end{aligned} \tag{17}$$

$$\begin{aligned}
\frac{d\widetilde{\gamma}}{dt} = {}& 6\pi\nu(2\pi\nu M)^{2/3}(1-e^2)^{-1}\left[1+\frac{1}{4}(2\pi\nu M)^{2/3}(1-e^2)^{-1}(26-15e^2)\right] \\
&-12\pi\nu\cos\lambda(S/M^2)(2\pi M\nu)(1-e^2)^{-3/2} \, ,
\end{aligned} \tag{18}$$

$$\begin{aligned}
\frac{de}{dt} = {}& -\frac{e}{15}(\mu/M^2)(1-e^2)^{-7/2}(2\pi M\nu)^{8/3}\big[(304+121e^2)(1-e^2)\big(1+12(2\pi M\nu)^{2/3}\big) \\
&-\frac{1}{56}(2\pi M\nu)^{2/3}\big((8)(16{,}705)+(12)(9082)e^2-25{,}211e^4\big)\big] \\
&+e(\mu/M^2)(S/M^2)\cos\lambda\,(2\pi M\nu)^{11/3}(1-e^2)^{-4}\big[(1364/5)+(5032/15)e^2 \\
&+(263/10)e^4\big] \, ,
\end{aligned} \tag{19}$$

$$\frac{d\alpha}{dt} = 4\pi\nu(S/M^2)(2\pi M\nu)(1-e^2)^{-3/2} \, . \tag{20}$$

It is computationally expensive to solve the ODEs using a time interval of 15 s, which corresponds to the observational cadence of LISA. However, the slow evolution of the orbital parameters predicted for most EMRI sources allows us to use a larger cadence of 15,360 s when solving the ODEs. As suggested in [81], the fifth-order Cash-Karp Runge–Kutta ODEs solver [83] is used at the larger cadence, and the solutions are then interpolated to the desired cadence of 15 seconds.

With the ODEs solutions at our disposal, we can now proceed to the calculation of the polarized waveforms. For each harmonic labeled as $(n, 2, m)$, the following quantities in their polarized waveforms are time independent: (1) the amplitude factor A such as $1/D$, (2) the initial phase $\Phi_0^{n2m} = n\phi_0 + 2\widetilde{\gamma}_0 + m\alpha_0$, and (3) the time-independent amplitude $\mathcal{A}_{+,\times}^{c,m}(\theta_s, \phi_s, \lambda, \theta_k, \phi_k)$. The exact forms of $\mathcal{A}_{+,\times}^{c,m}(\theta_s, \phi_s, \lambda, \theta_k, \phi_k)$ are provided in [81], and the superscript $c$ indicates that the quantity is an unknown constant. Therefore, the polarized waveforms can be expressed as follows:

$$\overline{h}_{+,\times}^{n2m}(\Theta) = \mathrm{A}\,\overline{s}_{+,\times}^{n2m}(\theta') = \mathrm{A}\,\mathrm{Re}\big(e^{i\Phi_0^{n2m}}\mathcal{A}_{+,\times}^{c,m}(\theta_s, \phi_s, \lambda, \theta_k, \phi_k)\overline{x}^n(\theta'')\big) \, , \tag{21}$$

where the parameter set $\Theta$ contains 14 parameters; $\theta'$ denotes the 13 parameters excluding $D$; $\theta$ represents the 8 parameters excluding $D$, $\phi_0$, $\widetilde{\gamma}_0$, $\alpha_0$, $\theta_k$, and $\phi_k$; and the parameter set $\theta''$ includes the 6 ODE-related parameters, $\mu$, $M$, $\lambda$, $S/M^2$, $e_0$, and $\nu_0$. Thus, we have

$$\Theta = \theta' \cup \{D\}, \theta' = \theta \cup \{\phi_0, \widetilde{\gamma}_0, \alpha_0, \theta_k, \phi_k\}, \theta = \theta'' \cup \{\theta_s, \phi_s\}. \tag{22}$$

Based on the number of parameters that they depend on, $\overline{h}_{+,\times}^{n2m}(\Theta)$ and $\overline{s}_{+,\times}^{n2m}(\theta')$ denote the 14 and 13-dimensional polarized waveforms, respectively, while the time-varying components correspond to the term $\overline{x}^n(\theta'')$, where the power distributions among harmonics depend on the index $n$. In the case of the AK model, the range of values for $m$ is from $-2$ to 2, resulting in a total of 5 harmonics for each $n$. Here, we adopt the same choice as our previous work [54] to select the loudest 10 harmonics by analyzing $\overline{x}^n(\theta'')$. Therefore, the choice is to pick up two values for $n$ from the values 1, 2, 3, 4, 5 used in LDC. It is worth mentioning that additional harmonics could be considered once computational limitations, such as accessing sufficient cores or utilizing a Graphics Processing Units (GPUs) code, are

overcome. However, for the current study, we focus on the loudest 10 harmonics based on the cluster resources available to us. As shown in Table 1, the power distributions among harmonics indicate that the harmonics with $n = 1$ are considerably weaker compared to other harmonics with different values of $n$. Furthermore, as $n$ increases (with $n \geq 2$), the strength of the harmonics diminishes. This trend holds true for moderately eccentric sources, such as those with $e_0 \leq 0.5$. Table 1 follows the same conventions as Table 1 in our previous paper [54], with two exceptions: (1) the harmonics indices become $n$, varying from 1 to 5, and (2) the power fraction is used instead of the SNR fraction, as their summation equals unity. Therefore, for moderately eccentric sources, the optimal choice for the loudest 10 harmonics would be those with $n \in \{2, 3\}$. It requires more attention to select the dominant harmonics for high eccentric sources, e.g., $e_0 > 0.5$, where the power distributions across harmonics exhibit greater fluctuations.

**Table 1.** Illustration of variation in the order of contributions of harmonics to the total power of an EMRI signal as a function of its parameters. Five typical signals are represented in the format of $C/F$, where $C$ indicates the harmonics index, and $F$ represents the corresponding power fraction of the signal. The row labeled as "power fraction" indicates the cumulative power of the top 10 harmonics. Signals in columns 3, 4, 5, and 6 utilize the LDC-1.2 [81] parameters but modify only one parameter specified in the header row.

| SNR Order (Descending) | LDC-1.2 Parameters | $\mu = 10 M_\odot$ | $\mu = 100 M_\odot$ | $e_0 = 0.5$ | $e_0 = 0.6$ |
|:---:|:---:|:---:|:---:|:---:|:---:|
| 1 | 2/0.654 | 2/0.583 | 2/0.855 | 2/0.362 | 4/0.338 |
| 2 | 3/0.281 | 3/0.326 | 3/0.123 | 3/0.338 | 5/0.334 |
| power fraction | 0.935 | 0.909 | 0.978 | 0.700 | 0.671 |
| 3 | 4/0.053 | 4/0.075 | 4/0.015 | 4/0.184 | 3/0.241 |
| 4 | 5/0.007 | 5/0.012 | 1/0.005 | 5/0.085 | 2/0.059 |
| 5 | 1/0.005 | 1/0.005 | 5/0.002 | 1/0.031 | 1/0.029 |

## 3. Generalized Likelihood Ratio Test

### 3.1. 13-Dimensional LLR

In the context of stationary Gaussian noise, the log-likelihood ratio (LLR) of given data $\overline{d}^I$ containing an assumed EMRI signal $\overline{h}^I(\Theta)$ is defined as follows:

$$\Lambda(\Theta) = \sum_{I \in \{A,E\}} \left[ -(\overline{h}^I(\Theta)|\overline{h}^I(\Theta)) + 2(\overline{d}^I|\overline{h}^I(\Theta)) \right] . \tag{23}$$

The $\overline{h}^I(\Theta)$ is usually called *template* in matched filtering to distinguish it from the unknown and true signal encoded in the noisy data. In the Generalized Likelihood Ratio Test [46], the global maximum of the LLR $L_G$ and the corresponding location $\widehat{\Theta}$, where

$$L_G = \Lambda(\widehat{\Theta}) , \tag{24}$$

$$\widehat{\Theta} = \underset{\Theta}{\mathrm{argmax}}\, \Lambda(\Theta) , \tag{25}$$

are used for signal detection and parameter estimation, respectively. Analytically maximizing over A by $\partial \Lambda(\theta', A)/\partial A = 0$ leads to

$$L_G = \max_{\theta'} \rho(\theta') , \tag{26}$$

$$\rho(\theta') = \max_{A} \Lambda(\Theta) = \frac{\left[ \sum_{I \in \{A,E\}} (\overline{d}^I|\overline{s}^I(\theta')) \right]^2}{\left[ \sum_{I \in \{A,E\}} (\overline{s}^I(\theta')|\overline{s}^I(\theta')) \right]} , \tag{27}$$

with the maximizer being

$$\widehat{A} = \underset{A}{\operatorname{argmax}} \, \Lambda(\Theta) = \frac{\left[ \sum_{I \in \{A,E\}} (\overline{d}^I | \overline{s}^I(\theta')) \right]}{\left[ \sum_{I \in \{A,E\}} (\overline{s}^I(\theta') | \overline{s}^I(\theta')) \right]} \, . \tag{28}$$

We call $\rho(\theta')$ the 13-dimensional LLR [44]. Creating further nested levels in the maximization of $\rho(\theta')$ that separate out the time-independent parts provide reduced dimensionality LLRs, namely, 8-dimensional and 7-dimensional ones, as discussed below.

*3.2. 8-Dimensional LLR*

By incorporating the polarized waveforms of the *i*-th harmonic in Equation (21) with the antenna patterns of arm *l* in Equation (1), we can obtain the corresponding strain response as follows:

$$\begin{aligned}
\overline{s}_l^i(\theta') =& F_l^+(\theta_s, \phi_s, \psi) \overline{s}_+^i(\theta') + F_l^\times(\theta_s, \phi_s, \psi) \overline{s}_\times^i(\theta') \, , \\
=& \mathrm{Re}(e^{i\Phi_0^i} \mathcal{A}_+^c(\theta_k, \phi_k, \theta_s, \phi_s, \lambda)) F_l^+(\theta_s, \phi_s, \psi) \mathrm{Re}(x^i(\theta'')) \\
&- \mathrm{Im}(e^{i\Phi_0^i} \mathcal{A}_+^c(\theta_k, \phi_k, \theta_s, \phi_s, \lambda)) F_l^+(\theta_s, \phi_s, \psi) \mathrm{Im}(x^i(\theta'')) \\
&+ \mathrm{Re}(e^{i\Phi_0^i} \mathcal{A}_\times^c(\theta_k, \phi_k, \theta_s, \phi_s, \lambda)) F_l^\times(\theta_s, \phi_s, \psi) \mathrm{Re}(x^i(\theta'')) \\
&- \mathrm{Im}(e^{i\Phi_0^i} \mathcal{A}_\times^c(\theta_k, \phi_k, \theta_s, \phi_s, \lambda)) F_l^\times(\theta_s, \phi_s, \psi) \mathrm{Im}(x^i(\theta'')), \\
=& \sum_{p=1}^{4} a_p^i \overline{x}_{l,p}^i(\theta) \, .
\end{aligned} \tag{29}$$

Here, the map for harmonics indices from $(n, m)$ to $i$ are $n = \lfloor (i-1)/5 \rfloor + 1$ and $m = ((i-1) \bmod 5) - 2$, where $i$ ranges from 1 to 25 in LDC. The linearity from strain responses to TDI responses for combination *I* leads to the same linear combination,

$$\overline{s}^{I,i}(\theta') \quad = \quad \sum_{p=1}^{4} a_p^i \overline{x}_p^{I,i}(\theta) \, , \tag{30}$$

because only the time-varying terms $\overline{x}_p^{I,i}(\theta)$ are projected to the TDI delays, and the time-independent coefficients $a_p^i$, which absorb the parameters $\phi_0$, $\widetilde{\gamma}_0$, $\alpha_0$, $\theta_k$, and $\phi_k$, remain unchanged.

To apply this linear decomposition in Equation (30) to the inner products in the 13-dimensional LLR in Equation (27), we can express the inner products as follows:

$$\begin{aligned}
(\overline{d}^I(\theta') | \overline{s}^I(\theta')) &= \sum_{i=1}^{N} \sum_{p=1}^{4} a_p^i (\overline{d}^I | \overline{x}_p^{I,i}(\theta)) \, , \\
(\overline{s}^I(\theta') | \overline{s}^I(\theta')) &= \sum_{i=1}^{N} \sum_{j=1}^{N} \sum_{p=1}^{4} \sum_{q=1}^{4} a_p^i a_q^j (\overline{x}_p^{I,i}(\theta) | \overline{x}_q^{I,j}(\theta)) \, .
\end{aligned} \tag{31}$$

In our previous work [54], we introduced an approach in which the three initial angles $\phi_0$, $\widetilde{\gamma}_0$, and $\alpha_0$ are separated from the remaining 10 parameters in Equation (27). This allows us to apply local maximization [84] over the three initial angles for a given point in the 10-dimensional parameter space and perform the search over the 10 parameters using PSO. In this paper, we extend the approach by employing local maximization [84] over the five parameters: $\theta_k$, $\phi_k$, $\phi_0$, $\widetilde{\gamma}_0$, and $\alpha_0$, using PSO for the remaining 8-dimensional search. The following quantities, $(\overline{d}^I | \overline{x}_p^{I,i}(\theta))$ and $(\overline{x}_p^{I,i}(\theta) | \overline{x}_q^{I,j}(\theta))$ can be pre-calculated for each specific $\theta$. This enables computationally efficient local maximization over the coefficients $a_p^i$, namely, over $\theta_k$, $\phi_k$, $\phi_0$, $\widetilde{\gamma}_0$, and $\alpha_0$.

The nature of the fitness function over the 5-dimensional subspace, consisting of $\theta_k$, $\phi_k$, $\phi_0$, $\widetilde{\gamma}_0$, and $\alpha_0$, is illustrated in Figure 1. The figure showcases the LLR (square root) landscape of a 2-dimensional slice of $\theta_k$ and $\phi_k$ (Figure 1a), as well as three randomly selected planes (Figure 1b–d) in the 3-dimensional subspace composed of $\phi_0$, $\widetilde{\gamma}_0$, and $\alpha_0$. This representation is valid for the specific location, although similar patterns are observed from the other locations as well. Given the presence of a fairly small number of local maxima with comparable or equal values, the local maximization approach is well-suited for handling this 5-dimensional subspace. To ensure that the global maximum is caught, we employed a total of 243 independent runs of a local maximizer starting from initial points distributed over a grid, with each angle in the 5-dimensional subspace enumerated from the 1-dimensional grid $\{0, 2\pi/3, 4\pi/3\}$, which are uniform spacings from 0 to $2\pi$. The best-fit 5-dimensional location is determined from the run that returns the highest value.

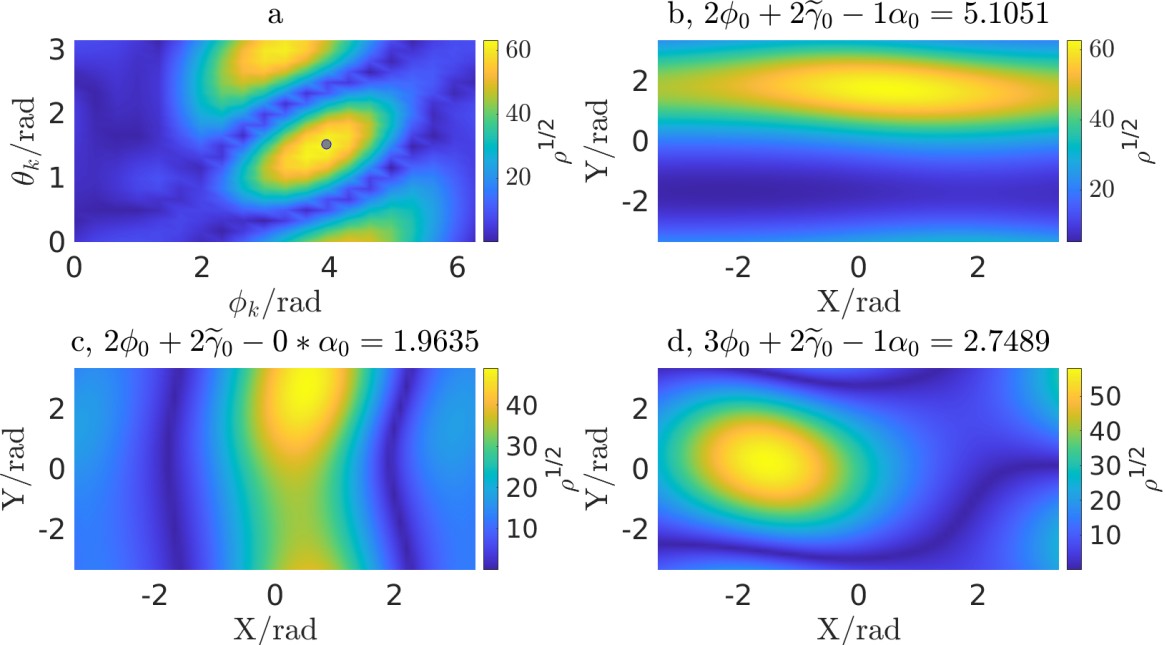

**Figure 1.** Illustrations of structures of the 5-dimensional subspace evaluated at a location. (**b–d**) The $X$ and $Y$ axes lie in these planes in the 3-dimensional subspace composed of $\phi_0$, $\widetilde{\gamma}_0$, and $\alpha_0$, and the range along both is $[-\pi, \pi]$. (**a**) The dot in panel a denotes the true values of the corresponding locations of the injected signal, which also labels the global maximum of the given landscape.

### 3.3. 7-Dimensional LLR

The initial orbital frequency, namely, $\nu_0$, corresponds to the moment $t_0$ at which the EMRI signal is captured by the detector, thus its varying results in a uniform shift of time labels to all the harmonics of the signal. As discussed in [47], the corresponding shift of the time label can be numerically maximized in two ways for an arbitrary harmonic, denoted as $\overline{x}$ here. The first is a phase rotation in the frequency domain:

$$\overline{x}(t - n\Delta t) = \frac{1}{N} \sum_{k=0}^{N-1} \widetilde{x}(f_k)e^{-i2\pi f_k(t-n\Delta t)} = \frac{1}{N} \sum_{k=0}^{N-1} [\widetilde{x}(f_k)e^{i2\pi f_k n\Delta t}]e^{-i2\pi f_k t} \,, \tag{32}$$

where $n$ denotes the number of the shift and $\Delta t$ represents the observational cadence. The inverse Fast Fourier Transform of the term $\widetilde{x}(f_k)e^{i2\pi f_k n\Delta t}$, which rotates the $\widetilde{x}(f_k)$ by the same amount of $n\Delta t$ at each $f_k$, returns the delayed term $\overline{x}(t - n\Delta t)$. For the same shift, the second is a straightforward lag sliding in the time domain as follows:

$$(x_0, x_1, \ldots, x_{N-1}) \xrightarrow{n} (x_n, x_{n+1}, \ldots, x_{N-1}, 0, \ldots, 0) \,, \tag{33}$$

where the zero paddings at the end of the shifted signal cover $n$ zeros.

The detector noise in the low-frequency region is usually large; as a result, a fiducial $\nu_0$, e.g., 1 mHz, can be determined through a pre-analysis of the detector's features, which indicates that the detector has reached a level of sensitivity to detect the GWs of EMRI signals starting from the chosen fiducial $\nu_0$. Therefore, the 8-dimensional TDI responses $\overline{x}_p^{I,i}(\theta)$ in Equation (30) could be calculated by running forward ODEs using $\theta$ with its initial $\nu_0$ specified as the selected fiducial value, and the initial $e_0$ being one of the parameters for matched filtering. We can then systematically shift the $\overline{x}_p^{I,i}(\theta)$ lag-by-lag starting from the lag of the fiducial $\nu_0$ until the 8-dimensional LLR maximum is achieved. The corresponding lag provides the best-fit estimation of $e_0$ and $\nu_0$. Here, we set the number of shifts to 11 for computational limitations.

Figure 2 illustrates the 8-dimensional and the 7-dimensional LLRs which share the same values for parameter set $\theta \setminus \{\nu_0\}$ but vary $\nu_0$ for the former and the fiducial $\nu_0$ for the latter. The lag varies from $-10$ to 10 where the zero lag corresponds to the true lag of LDC $\nu_0$, $7.3804631408 \times 10^{-4}$ Hz. It can be observed that the 8-dimensional LLRs using the negative lags can be successfully mapped to the 7-dimensional LLRs with a well-fitted $\nu_0$ by properly shifting the corresponding $\overline{x}_p^{I,i}(\theta)$. This is possible because the total 11 shifts can cover the zero lag anyway, whereas the positive lags fail to locate the zero lag due to the rightward shift of $\overline{x}_p^{I,i}(\theta)$.

In this paper, the lag of the fiducial $\nu_0$ is determined by considering 4 lags ahead of the LDC's true lag; thus, the corresponding value is $7.3804587134 \times 10^{-4}$ Hz. In order to accurately capture unknown EMRI signals, it would generally be necessary to fit more lags. However, due to the computational expense of the shifting operations for $\overline{x}_p^{I,i}(\theta)$ and the evaluations of the 8-dimensional LLRs by using the current code, only 11 lag-by-lag shifts are utilized, enumerating lags from $-4$ to 6 as illustrated in Figure 2. This setting ensures the scanning of the true lag, and it is used to demonstrate the functionality of the 7-dimensional LLR. In future works, we plan to address these computational challenges by implementing a GPU-accelerated code, which will allow for the exploration of additional lags.

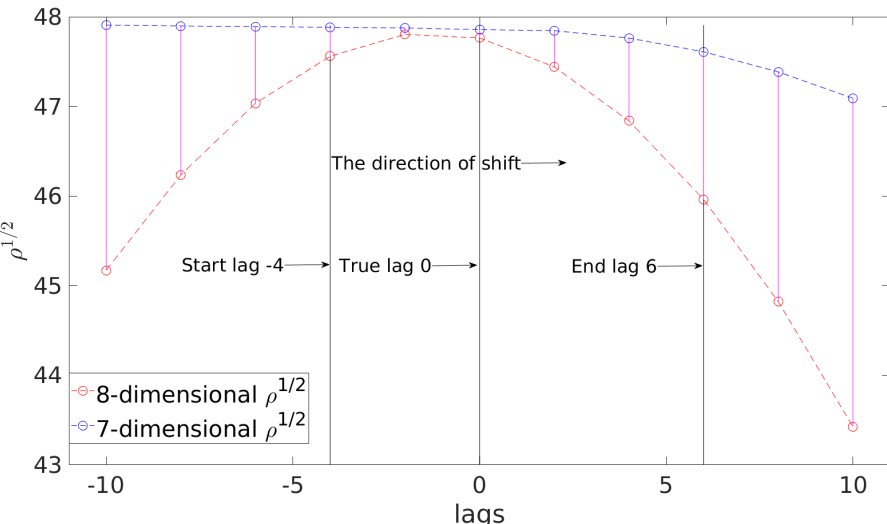

**Figure 2.** Illustrations of the square root of the LLRs over lags. The square roots of the 8-dimensional LLRs are in red and the corresponding 7-dimensional values are in blue, connected with a solid magenta line for each lag.

## 4. Particle Swarm Optimization

As discussed earlier, the search using reduced dimensional likelihoods involves the following steps. First, the distance $D$ in the 14-dimensional LLR in Equation (23) is analytically maximized. Next, the local maximization over the five angles $\theta_k$, $\phi_k$, $\phi_0$, $\widetilde{\gamma}_0$, and $\alpha_0$ is carried out using the Simplex algorithm of Nelder and Mead [84]. Finally, the

remaining parameters in the set $\theta$ (8-dimensional search), or $\theta$ excluding $\nu_0$ (7-dimensional search), are numerically maximized by PSO. In this chapter, we briefly describe the PSO algorithm [50,51,79].

Given the fitness function $f(\overline{x})$, where $\overline{x}$ is defined in $\mathbb{R}^M$, the optimization problem can be stated as follows:

$$\overline{x}_* = \underset{\overline{x} \in \mathbb{D} \subset \mathbb{R}^M}{\text{argmax}} f(\overline{x}) , \tag{34}$$

$$f(\overline{x}_*) \geq f(\overline{x}) , \forall \overline{x} \in \mathbb{D} . \tag{35}$$

The best location, $\overline{x}_*$, refers to the point in the search space $\mathbb{D}$ that yields the highest fitness value, represented as $f(\overline{x}_*)$; $M$ is the dimension of the parameter space for $f(\overline{x})$. Locating the primary peak of a multimodal fitness function can be challenging. The PSO algorithm, which is utilized in this paper as a global maximizer, is a suitable approach to addressing such challenges. Successful applications of PSO in handling similar issues are discussed in Section 1. It should be noted that in our case, the fitness functions are the 8-dimensional LLR discussed in Section 3.2 and the 7-dimensional LLR discussed in Section 3.3.

PSO consists of multiple agents known as particles. Each particle updates its position by considering the information from both itself and its neighboring particles at each iteration. The algorithm aims to converge towards the global maximum, which corresponds to the primary peak of the fitness function within the search space, by utilizing a balance between global exploration and local exploitation. Such balance typically results in the good performance of a PSO search. However, finding the right balance requires tuning the related parameters, which is problem-specific. One of the key advantages of the PSO algorithm is that it requires only a few tunable parameters, namely, the number of iterations $N_{\text{iter}}$ and the number of independent runs $N_{\text{runs}}$ of PSO. If the probability that an individual PSO fails to locate the primary peak of the fitness function is denoted as $p$, then the probability that at least one search from $N_{\text{runs}}$ independent PSO searches, using different random seeds, succeeds in locating the primary peak is given by $1 - p^{N_{\text{runs}}}$. This probability approaches unity exponentially fast with $N_{\text{runs}}$. Therefore, multiple independent runs are a quick and easy way to significantly enhance the performance of a PSO-based search. It is recommended to start with $N_{\text{runs}}$ in the range of 6~12 and to set $N_{\text{iter}}$ to 2000, as discussed in [79]. These values can be adjusted based on the specific fitness function being used. The actual values used in this paper are described in Section 5. For more detailed information on an objective strategy for tuning PSO parameters, refer to [55].

The PSO dynamics of the $i$-th particle in the swarm is described by two equations as follows:

$$\overline{x}_i(t+1) = \overline{x}_i(t) + \overline{v}_i(t+1) , \tag{36}$$

$$v_i^j(t+1) = w v_i^j(t) + c_1 r_1 (p_i^j(t) - x_i^j(t)) + c_2 r_2 (g_j(t) - x_i^j(t)) , \tag{37}$$

where $t$ represents an iteration, and $\overline{x}_i(t)$ and $\overline{x}_i(t+1)$ denote the respective positions before and after the update. $\overline{v}_i(t+1)$ represents the amount of positional increment, referred to as velocity, while $v_i^j(t+1)$ is the corresponding projection component for the $j$-th parameter. The quantities $x_i^j(t)$ and $p_i^j(t)$ represent the current location and *personal best* (*pbest*) location of the $j$-th parameter, while $g_j(t)$ represents the *global best* (*gbest*) location among all particles of the $j$-th parameter. Equation (37) provides the key feature of a PSO update. The first term represents the influence of the momentum of the $i$-th particle with $\omega$ being the inertia weight. The second and third terms represent the acceleration effects, where the former considers the influence of the particle itself and the latter represents the influence from neighboring particles, with $c_1$ and $c_2$ being the acceleration coefficients. The randomness of the PSO algorithm arises from the utilization of random variables $r_1$ and $r_2$ , which are drawn from a uniform distribution between 0 and 1. The locations of *pbest* and *gbest* are updated following the rules below:

$$\text{if } f(\overline{x}_i(t)) > f(\overline{p}_i(t)), \text{ then } \overline{p}_i(t+1) = \overline{x}_i(t+1) \,, \tag{38}$$

$$\text{if } f(\overline{x}_i(t)) > f(\overline{g}(t)), \text{ then } \overline{g}(t+1) = \overline{x}_i(t+1) \,. \tag{39}$$

The typical settings for PSO are as follows: (1) $c_1 = c_2 = 2$; (2) linearly decreasing inertia weight $\omega$ over iterations; (3) constraining the velocity by a given parameter, referred to as the maximum velocity, $V_{\max}$, whose value is usually 0.2, but 0.5 is used here to strengthen the search ability of PSO, such that $-V_{\max} \le v_i^j(t) \le V_{\max}$ for all iterations and particles; (4) randomly generating initial positions and velocities for all particles; and (5) setting the number of particles $N_p$ in the swarm to $N_p = 40$. The "let-them-fly" boundary condition is used, where the position and velocity of a particle remain unchanged, and a fitness value of $-\infty$ is assigned once the particle leaves the search space. As a result, the actual number of fitness function (likelihood) evaluations for an individual PSO search would be smaller than the value of $N_{\text{iter}} \cdot N_p$.

To enhance the exploitation capability of PSO, particularly for multimodal fitness functions, a variation called *local best* (*lbest*) PSO [51] is proposed as an improvement over the *gbest* PSO. In the *lbest* PSO, for each particle $i$, a smaller swarm is utilized to determine the *lbest* position denoted as $\overline{p}_{\text{local},i}(t)$ and the corresponding fitness value $f(\overline{p}_{\text{local},i}(t))$. These values are then used to replace the *gbest* position $g_j(t)$, $\overline{g}(t+1)$ in Equation (37) and the corresponding fitness value $f(\overline{g}(t))$ in Equation (39). The typical configuration for the smaller swarm surrounding the $i$-th particle is a ring structure consisting of three particles, whose indices are given by $\mathcal{N}_i = i - 1, i, i + 1$, with the first and last particle connected in a circular manner. It is worth noting that the *lbest* PSO reduces to the *gbest* PSO when the ring includes all the particles. The selection of the fitness value for the *lbest* of the $i$-th particle follows the criteria shown below:

$$f(\overline{p}_{\text{local},i}(t)) = \max_{j \in \mathcal{N}_i} f(\overline{p}_j(t)) \,. \tag{40}$$

Here, a more comprehensive exploitation is achieved by slower convergence in the *lbest* PSO, thus making it more computationally expensive than *gbest* PSO.

## 5. Results

In this study, we utilized 0.5 years of data containing a single EMRI signal with the same source parameters as LDC-1.2 [81], except for a shorter distance $D$ of 1.535300 Gpc, resulting in an SNR value of 50 for the injected signal. This SNR value has been widely used as a benchmark for 0.5 years signals in recent studies [30,31,43]. While our search method is not inherently restricted to a shorter data length, constraints on computational resources and a pending GPU-acceleration of our code sets the above limit on the data length. The noise realization used in our analysis is obtained by subtracting the signal from the data, with both provided in LDC-1.2 [81], ensuring that our simulated data share the same characteristics as the LDC data but with a scaling of a shorter duration and a higher SNR. In Figure 3, the spectra of the injected signal and the simulated data for TDI combinations $A$ and $E$ are displayed, revealing the relatively weak nature of the injected signal compared to the simulated data. The values of the source parameters and the width of the search ranges used for the 8-dimensional and the 7-dimensional searches are presented in Table 2. The Fisher Information Matrix (FIM) $\sigma$ represents the estimation error of the Cramer–Rao Lower Bound (CRLB) for each parameter at an SNR of 50, evaluated at the injected source parameters. The injected signal parameters are also called the true ones in the following analysis. We set the tunable hyperparameters for PSO as follows: $N_{\text{runs}}$ is set to 6, and $N_{\text{iter}}$ is set to 15,000 for the 8-dimensional searches, 20,000 for the first two 7-dimensional searches, and 25,000 for the remaining four 7-dimensional searches. Due to limited computational resources, the 6 independent searches have to be carried out serially. Due to the presence of noise in the data, both PSO and local maximization are expected to find best-fit fitness values that are higher than that at the true location, which

are called a successful search. In order to further reduce computational costs, the searches are terminated once a successful search occurs. Consequently, the actual $N_{\text{runs}}$ is 4 for the 8-dimensional searches and 6 for the 7-dimensional searches.

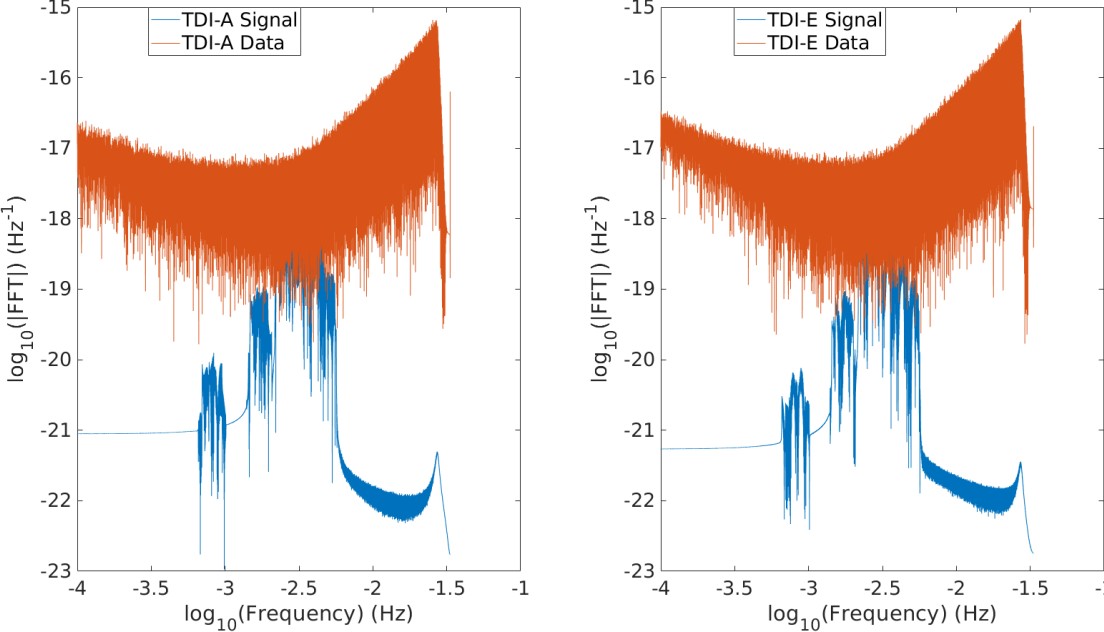

**Figure 3.** Magnitudes of the FFTs of the injected signal with SNR = 50 in blue and the corresponding data in red, where the TDI combination $A$ is illustrated in the left panel and the TDI combination $E$ is displayed in the right panel. See their definitions in Equation (7).

**Table 2.** The injected source parameters and range width used in our search. Currently, the location of the injected signal is set at the center of the given range. We leave a more general search, with injected signals placed non-centrally in the search space, to future work. The value of 0.3456 (7D) is the corresponding orbital frequency difference between lag 6 and lag −4 relative to its $\sigma$ value; see more details in Section 3.3.

| Parameters | LDC Values | FIM $\sigma$ | Search Range Absolute Value | Search Range in $\sigma$ |
|---|---|---|---|---|
| $\mu(M_\odot)$ | 29.490000 | $4.872139 \times 10^{-2}$ | 1 | 20.5249 |
| $M(M_\odot)$ | $1.1349449 \times 10^{6}$ | $3.582834 \times 10^{3}$ | $10^{5}$ | 27.9109 |
| $\lambda(\text{rad})$ | 2.1422000 | $9.471417 \times 10^{-3}$ | $\pi/16$ | 20.7307 |
| $S/M^2$ | 0.9697 | $3.15374 \times 10^{-3}$ | 0.1 | 31.7084 |
| $e_0$ | 0.22865665 | $1.842612 \times 10^{-4}$ | 0.005 | 27.1354 |
| $\nu_0(\text{Hz})$ | $7.3804631 \times 10^{-4}$ | $3.202842 \times 10^{-9}$ | $3.202842 \times 10^{-7}$ (8D) 11 lags (7D) | 100 (8D) 0.3456 (7D) |
| $\theta_s(\text{rad})$ | 0.4989445 | $2.415649 \times 10^{-3}$ | $\pi$ | 1300.5 |
| $\phi_s(\text{rad})$ | 2.232797 | $1.708559 \times 10^{-3}$ | $2\pi$ | 3677.5 |

The results obtained from the 8-dimensional and the 7-dimensional searches are summarized in Table 3 and Table 4, respectively. We report the square roots of the best-fit fitness values from each PSO search, which provide the estimated SNRs. Additional details regarding Tables 3 and 4 are provided below.

1. The 4-th PSO in the 8-dimensional searches is successful as indicated by the estimated SNR shown in bold. However, no similar successful search is observed in the 7-dimensional searches.

2. Parameter estimation errors are determined by subtracting the corresponding signal parameter's best-fit values from their true values. The six ODE-related parameters, namely, $\mu$, $M$, $\lambda$, $S/M^2$, $e_0$, and $\nu_0$, are expressed relative to their respective FIM $\sigma$ (evaluated at the true location). The estimation error for $D$ is expressed relative to its true value itself. For the parameters $\theta_s$ and $\phi_s$ that represent the sky's location, we show the errors themselves. The sky's locations $(\theta_s, \phi_s)$ and $(\pi - \theta_s, \phi_s + \pi)$ [26] contribute a degeneracy to the LLR in Equation (27). As a result, we use the asterisk ($*$) to show the corresponding errors after the degeneracy is taken care of.

3. To consider the impact of weak harmonics beyond the loudest 10 on the estimation of the initial angles $\phi_0$, $\widetilde{\gamma}_0$ and $\alpha_0$, as well as the angles $\theta_k$ and $\phi_k$ denoting the spin direction of the MBH, we conduct a rerun of the 5-dimensional local maximization using a waveform with all the 25 harmonics at the best-fit location from each PSO search, where the templates used in the search are restricted to the loudest 10 harmonics with $n \in \{2, 3\}$. The estimated angles are then utilized in the estimation of the distance $D$ using Equation (28).

4. The recovered 14-dimensional parameters obtained previously are utilized in Equation (6) and Equation (7) to estimate the signal of $A$ and $E$. The separate and combined overlaps between the injected and the estimated signal are quantified as $\mathrm{ff}_A$, $\mathrm{ff}_E$, and $\mathrm{ff}_{AE}$, respectively.

For the 8-dimensional PSO outputs shown in Table 3, it can be observed that the errors in the parameters $\mu$, $M$, $\lambda$, and $S/M^2$ are $\approx 2\sigma$, while the error in the parameter $D$ is $\approx 1\%$. The errors in the sky's location are within $\approx 0.1$ radians. However, the errors in the parameters $e_0$ and $\nu_0$ vary significantly among different PSO outputs where the successful PSO output returns a minimum error of approximately $\sim 2\sigma$ with an overlap of 98%, and the other PSO outputs yield larger errors up to $\sim 8\sigma$ with smaller overlap values. These discrepancies are reasonable because $e_0$ and $\nu_0$ are the initial values of the ODEs in Equation (20) which describe the orbital dynamics of the EMRI source (the other three initial angles do not determine the morphology of the ODEs' solution, only contributing a constant shift). Therefore, the phase match are more sensitive to these parameters, thus requiring longer iterations to converge.

For the 7-dimensional PSO outputs presented in Table 4, no successful search is found where the best-fit fitness value exceeds that at the true location. However, the 4th PSO output returns errors of approximately $\sim 1\sigma$ for the parameters $\mu$, $M$, $\lambda$, $S/M^2$, $e_0$, and $\nu_0$; $\sim 5\%$ for the distance $D$; and $\sim 5\%$ radians for the sky's location, with an overlap of 97%. This indicates that the signal is indeed captured, making it a successful search. The 3rd and 5th PSO outputs exhibit similar features to the first three PSO outputs in the 8-dimensional searches, where the larger errors in $e_0$ result in the smaller fitness values. The errors in $\nu_0$ are the same for the 1st, 2nd, 3rd, and 5th PSO outputs, which may be attributed to the small range of 11 lags used to shift the 8-dimensional $A$ and $E$ template starting from the lag of the fiducial $\nu_0$. It should be noted that the fitting of $\nu_0$ should cover more lags to obtain a more accurate estimation over $\nu_0$.

The successful PSO searches (the 4th PSO for both dimensions) demonstrate smaller errors in the parameters $\mu$, $M$, $\lambda$, $S/M^2$, $e_0$, and $\nu_0$ for the 7-dimensional search ($\sim 1\sigma$) compared to those for the 8-dimensional search ($\sim 2\sigma$). This suggests that the utilization of reduced dimensional LLR and increased iterations effectively reduce estimation errors, particularly for parameters that are related to the GW phase. The fact that all PSO runs obtained fitness values close to each other but at various offsets for estimated errors, ranging from $1\sigma$ to $8\sigma$, illustrates the presence of large number of secondary peaks in the fitness function.

**Table 3.** PSO outputs of 8-dimensional searches. The square root of the fitness value at the true 8-dimensional location is **47.879594**. Further details about the table are discussed in Section 5.

| | **1st PSO** | **2nd PSO** | **3rd PSO** | **4th PSO** |
|---|---|---|---|---|
| | | Square root of fitness values | | |
| Best location from PSO | 47.546001 | 46.381273 | 47.069351 | **47.988164** |
| | | Parameter estimation errors | | |
| $\mu(M_\odot)$ | −3.1 | −2.3 | 0.21 | 2.4 |
| $M(M_\odot)$ | 1.9 | 2.1 | −1.1 | −2.6 |
| $\lambda(\mathrm{rad})$ | −2.1 | −2.1 | 0.96 | 2.5 |
| $S/M^2$ | −2.2 | −2.2 | 0.91 | 2.5 |
| $e_0$ | 7.8 | 2.9 | 3.6 | −1.2 |
| $\nu_0(\mathrm{mHz})$ | −6.8 | −4.5 | −8.2 | −1.9 |
| $D(\mathrm{Gpc})$ | −0.03 | 0.00011 | −0.12521 | 0.015 |
| $\theta_s(\mathrm{rad})$ | 0.068 | −0.078970 * | 0.13 | −0.012 |
| $\phi_s(\mathrm{rad})$ | 0.015 | −0.167177 * | −0.0062 | 0.046 |
| | | Overlap between the estimated and true signals | | |
| ff$_A$ | −0.970817 | 0.972518 | 0.964058 | −0.990312 |
| ff$_E$ | −0.965563 | 0.940148 | 0.939171 | −0.982537 |
| ff$_{AE}$ | −0.968851 | 0.959972 | 0.954244 | −0.987405 |

**Table 4.** PSO outputs of 7-dimensional searches. The square root of the fitness value at the true 7-dimensional location is **47.882605**. Further details about the table are discussed in Section 5.

| | **1st PSO** | **2nd PSO** | **3rd PSO** | **4th PSO** | **5th PSO** | **6th PSO** |
|---|---|---|---|---|---|---|
| | | | Square root of fitness values | | | |
| Best location from PSO | 47.699082 | 47.329812 | 47.685694 | 47.738310 | 47.582240 | 47.023112 |
| | | | Parameter estimation errors | | | |
| $\mu(M_\odot)$ | 4.7 | 4.4 | 0.48 | −1.3 | −0.89 | 4.9 |
| $M(M_\odot)$ | −5.1 | −5.0 | −0.92 | 1.5 | 0.28 | −4.3 |
| $\lambda(\mathrm{rad})$ | 5.0 | 4.8 | 0.84 | −1.5 | −0.38 | 4.3 |
| $S/M^2$ | 5.0 | 4.8 | 0.82 | −1.5 | −0.4 | 4.3 |
| $e_0$ | −2.8 | −1.8 | 1.5 | 0.2 | 3.2 | −7.0 |
| $\nu_0(\mathrm{mHz})$ | −0.21 | −0.21 | −0.21 | −0.035 | −0.21 | 0.14 |
| $D(\mathrm{Gpc})$ | −0.09576 | −0.08430 | −0.04126 | 0.05260 | −0.05899 | −0.00204 |
| $\theta_s(\mathrm{rad})$ | 0.097603 * | 0.078 | 0.042 | −0.043020 * | 0.094 | −0.019956 * |
| $\phi_s(\mathrm{rad})$ | 0.006113 * | 0.06 | −0.048 | 0.050827 * | 0.039 | 0.091476 * |
| | | | Overlap between the estimated and true signals | | | |
| ff$_A$ | 0.977230 | 0.959595 | −0.976542 | −0.989005 | −0.969600 | −0.973063 |
| ff$_E$ | 0.966966 | 0.951818 | −0.969133 | −0.976612 | −0.958945 | −0.955183 |
| ff$_{AE}$ | 0.973175 | 0.956625 | −0.973700 | −0.984385 | −0.965498 | -0.966438 |

## 6. Discussion

We extended the previous work on a 10-dimensional LLR [54] search to an 8-dimensional and a 7-dimensional LLR search, in which progressively more parameters are locally maximized while the remaining are globally maximized using PSO. In the 8-dimensional search, we performed a 5-dimensional local maximization over the three initial angles $\phi_0$, $\widetilde{\gamma}_0$, and $\alpha_0$, and the angles $\theta_k$ and $\phi_k$ describing the spin direction of the MBH. In the 7-dimensional search, we used a fiducial value of $\nu_0$ and applied a lag-by-lag shift to the 8-dimensional TDI responses to fit the true $\nu_0$.

The low estimated errors and the corresponding high overlap between the estimated and injected signals indicate that both the 8-dimensional and the 7-dimensional search work well within a wider search range. Our approach used the same search range widths for $\mu$ and $M$ as the low mass-ratio sources prescribed in MLDC 1.3.4 and 1.3.5, and half the width of the MLDC value for the parameter $S/M^2$ [35]. This serves as a guide for future hierarchical searches, as demonstrated in [43] using certain clustering techniques, for how much they need to narrow down the search ranges for parameters such as $e_0$, $\nu_0$, and $\lambda$.

The larger errors observed for $e_0$ and $\nu_0$, compared to the smaller errors for other parameters in Tables 3 and 4, indicate that matched filtering is more sensitive to these two parameters. Thus, it becomes more difficult to accurately determine them. This insight inspires us to explore more advanced optimization algorithms, such as the Cooperative Coevolution Particle Swarm Optimization (CCPSO) [85], where only a subset of parameters are updated at each iteration to improve the optimization process. We expect that this approach will help PSO particles in escaping from secondary peaks and converging faster towards the primary peak in the parameter space of the fitness function. The additional computational cost can be mitigated by implementing a faster code using GPU acceleration.

In systems with higher eccentricity ($e_0 > 0.5$), the power distributions over harmonics become more erratic, which depend on the harmonics index $n$ only for the 8-dimensional waveform. Consequently, the loudest 10 harmonics, with fixed values $n$ belonging to the set $\{2, 3\}$, might not be the optimal choice any longer. Hence, we need to develop methods to select the dominant harmonics on the fly in such systems.

The existence of multiple secondary peaks can hinder the PSO update process, making it difficult for particles to converge towards the primary peak. As a result, larger estimation errors of the signal parameters may occur. To effectively tackle this issue, one possible approach is to employ the reduced dimensional LLR and increase the number of iterations for PSO searches. Nevertheless, the increased computational requirements necessitate the utilization of additional cores or GPUs in the code.

In our previous 10-dimensional searches, where computational costs were lower, we examined injected signals with SNR values of 50, 40, and 30 and a duration of 0.5 years. However, in this paper, our focus is on the computationally expensive 8-dimensional and 7-dimensional searches. Consequently, we only cover injected signals with an SNR of 50 and the same duration due to our limited computational resources. This SNR value of 50 is higher compared to the SNR of the LDC-1.2 signal with the same duration. In future work, it is important to explore lower SNR values to assess the robustness of our method. We also plan to conduct additional tests, such as a random placement of the true location, wider search ranges, and longer data duration, to further validate our approach.

**Author Contributions:** Conceptualization, S.D.M. and X.-B.Z.; software X.-B.Z. and S.D.M. (for PSO); validation X.-B.Z., S.D.M., H.-G.L. and Y.-X.L.; writing—original draft preparation, X.-B.Z.; writing—review and editing, S.D.M., H.-G.L. and Y.-X.L.; supervision, S.D.M. and H.-G.L.; project administration, Y.-X.L.; funding acquisition, H.-G.L. and Y.-X.L. All authors have read and agreed to the published version of the manuscript.

**Funding:** The work of X-B.Z., H-G.L. and Y-X.L. was supported by the following grants: (1) National Key Research and Development Program of China (Grant No. 2021YFC2203003); (2) National Key Research and Development Program of China (Grant No. 2021YFC2203002); (3) The National Key Research and Development Program of China (Grant No. 2022YFA1402704); (4) The National Natural Science Foundation of China (NSFC) (Grant No. 11834005 and No. 12247101).

**Data Availability Statement:** The LDC data used in this study are publicly available from https://lisa-ldc.lal.in2p3.fr(accessed on 3 April 2024). The PSO code used in this study is based on one provided in the GitHub repository RAAPTR available at https://github.com/yanwang2012/RAAPTR(accessed on 3 April 2024). The rest of the code simply implements the mathematical formalism for GLRT and MLE described in this paper, with all necessary details provided to aid reproducibility.

**Acknowledgments:** The computations were performed on the high-performance computers of the State Key Laboratory of Scientific and Engineering Computing, Chinese Academy of Sciences. We express our gratitude to RunQiu Liu for granting us access to the cluster and providing invaluable suggestions for our research. We also extend our appreciation to the administrator, Yin Qian, for assisting us in utilizing the cluster effectively. Furthermore, we would like to acknowledge the insightful discussions we had with Xian Chen, Wen-biao Han, Peng Xu, Wen-lin Tang, Qun-ying Xie, Xue-hao Zhang, Shao-dong Zhao, Yi-yang Guo, Han-zhi Wang, Shu-zhu Jin, Qian-yun Yun, and Hou-qiang Teng.

**Conflicts of Interest:** The authors declare no conflicts of interest. The funders had no role in the design of the study; in the collection, analyses, or interpretation of data; in the writing of the manuscript; or in the decision to publish the results.

## Abbreviations

The following abbreviations are used in this manuscript:

| | |
|---|---|
| AK | Analytical Kludge |
| CO | Compact Object |
| CRLB | Cramer–Rao Lower Bound |
| DFT | Discrete Fourier Transform |
| EMRI | Extreme Mass Ratio Inspiral |
| FIM | Fisher Information Matrix |
| GWs | Gravitational Waves |
| GLRT | Generalized Likelihood Ratio Test |
| GPUs | Graphics Processing Units |
| LDC | LISA Data Challenge |
| LLR | Log-Likelihood Ratio |
| LISA | Laser Interferometer Space Antenna |
| MCMC | Markov Chain Monte Carlo |
| MLDC | Mock LISA Data Challenge |
| MBH | Massive Black Hole |
| ODEs | Ordinary Differential Equations |
| PSD | Power Spectral Density |
| PSO | Particle Swarm Optimization |
| SNR | Signal-to-Noise Ratio |
| SSB | Solar System Barycenter |
| TDI | Time Delay Interferometry |

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
