# Peer review of "Search for Extreme Mass Ratio Inspirals Using Particle Swarm Optimization and Reduced Dimensionality Likelihoods"

_universe, doi:10.3390/universe10040171_

Round 1

Reviewer 1 Report

Comments and Suggestions for Authors

This work is an extension of the authors’s earlier ones which use a Particle Swarm Optimization method to search for EMRI signals in a high-dimensional, multiple-structured parameter space. The authors extended the search to an 8-dimensional one, and demonstrated the effectiveness of the scheme. The paper is written clearly.

In general, I find the results very impressive, and I would be happy to recommend this work for publication. I have only one comment regarding the search range, which I hope the authors can take into account in the revision. 

The search range is concentrated around the true values of the parameters, which, as I understand, is due to the limitation of computational resources. Ideally, the search range should cover the entire possible parameter space. This, of course, could be improved in future work, but I think some mentioning of the current limitation in the abstract would be helpful, so that readers can properly grasp the achievement of the current work. 

Reviewer 2 Report

Comments and Suggestions for Authors

Summary

This paper presents the results of an 8-dimensional and a 7-dimensional particle swarm optimization (PSO) search of extreme mass ratio inspirals. The paper extends the authors’ previous study using a 10-dimensional PSO. The authors provide a detailed method description and discuss the limitations of the methods and potential future work.

Suggestions

The paper will benefit from being more careful in the description and definition of the methods, as well as a discussion of how the 7-dimensional PSO does not observe a successful search of the 8-dimensional PSO: does the result indicate inherent problems with choosing a fiducial initial orbital frequency and using lag sliding, and does the benefits overweight the cost?

Comments

1. Do the authors mean “polarized waveform” whenever the phrase “polarization waveform
 is used? The phrase appears to describe “polarized waveforms”, and its use is not standard jargon to my knowledge.

2. Delay L is first used in line 120 while not defined until 126. Moreover, a potentially different L is used starting from line 137. The authors should define them clearly to avoid confusion.

3. Line 138: f is not used explicitly in the LISA Data Challenge manual [64]. If the authors are defining the quantity based on f=ω/2π, please explicitly note so to avoid confusion.

4. Line 224: use the mathematical symbol for the floor function.

5. Figure 1: what is the dot in Figure 1a? Does it represent the global maximum or the true value? The landscape illustrated in 1a appears to have potentially three peaks. Can the authors comment on why this is not a concern?

6. Figure 2: can the authors comment on how the square root of the log-likelihood ratio for the 7-dimensional case does not appear to vary significantly for lag -10 to 1, and if this has anything to do with the fact that there is no successful search for the 7-dimensional case?

7. Line 344: is there a value for Vmax used in this study?

8. Table 3 & 4: for the parameter estimation errors in terms of σ, it may be better to directly write out the values instead of in their scientific notation.

9. In the discussion, the authors mention “explore lower SNR values” in the future. Since in the author’s previous paper, SNR 30, 40, and 50 are already explored for the 10-dimensional PSO, it is helpful to be more specific in this statement.

Comments on the Quality of English Language

1. Even though the authors listed the abbreviations used in the paper at the end, it is still helpful to define them at their first appearances: MBH (line 6), DFT (line 144), FIM (line 378). CRLB (line 378)

2. Line 178, 256: it’s -> its?

3. Line 199: moderate eccentric-> moderately eccentric?

4. Line 201” the harmonics indices “turn to” -> become?

5. Line 263: the second is “a” straightforward lag sliding in “the” time domain…

Reviewer 3 Report

Comments and Suggestions for Authors

It is an interesting and well-organized paper.  EMRIs are gravitational wave sources for space-based detectors and it can last for months to years. Considering the long duration of EMRIs, it is important to develop a computational efficient algorithm to search for such signals.  The authors extended their previous study, and proposed an 8-dimensional PSO and a 7-dimensional PSO search. After reducing the parameter dimension of the likelihoods, it enables ones to search for 0.5-year EMRI signals within a wider range, and it provides guidelines for the future hierarchical searches. I recommend this paper to be published after addressing the following comments:

  1. In the introduction, please add a few more sentences to discussed the scientific importance of EMRI and cite more related papers. Besides, please add some a few sentences about the event rate of EMRIs.

  1. In the introduction, please add more discussion about the PSO method, like what are the advantages and disadvantages (or limitations) of this method. 

  1. The authors used the AK waveform to generate signals. It could be more convenient to use FEW package to generate more complicate waveform models: https://bhptoolkit.org/FastEMRIWaveforms/html/index.html . I suggest the authors to use this package in the future work.

  1. In both Table 3 and Table 4, is there any direct explanation for why the values of the overlap “ffE” are always smaller than those of “ffA”? 

  1. To improve the readability, please use a few sentences to explain the conventions of Table 1, instead of referring to Ref. [44].

Round 2

Reviewer 3 Report

Comments and Suggestions for Authors

I think the authors have addressed all my concerns. I recommend the paper to be published.

Author Response

Dear Reviewer,

Thanks for your positive response.

We are taking efforts to push the work further developed by absorbing the suggestions from all  of you.

Best Regards,

Xiao~Bo Zou